# Alterations of the Gut Microbiome in Chronic Hepatitis B Virus Infection Associated with Alanine Aminotransferase Level

**DOI:** 10.3390/jcm8020173

**Published:** 2019-02-02

**Authors:** Yeojun Yun, Yoosoo Chang, Han-Na Kim, Seungho Ryu, Min-Jung Kwon, Yong Kyun Cho, Hyung-Lae Kim, Hae Suk Cheong, Eun-Jeong Joo

**Affiliations:** 1Gwanghwamun Medical Study Centre, Syntekabio Inc., Seoul 03186, Korea; yeojuny@syntekabio.com; 2Department of Occupational and Environmental Medicine, Kangbuk Samsung Hospital, Sungkyunkwan University School of Medicine, Seoul 04514, Korea; yoosoo.chang@gmail.com (Y.C.); sh703.yoo@gmail.com (S.R.); 3Center for Cohort Studies, Total Healthcare Center, Kangbuk Samsung Hospital, Sungkyunkwan University, School of Medicine, Seoul 04514, Korea; hanna147942@gmail.com (H.-N.K.); mjkkmd@gmail.com (M.-J.K.); 4Department of Clinical Research Design and Evaluation, SAIHST, Sungkyunkwan University, Seoul 06351, Korea; 5Medical Research Institute, Kangbuk Samsung Hospital, Sungkyunkwan University, School of Medicine, Seoul 03181, Korea; 6Department of Laboratory Medicine, Kangbuk Samsung Hospital, Sungkyunkwan University School of Medicine, Seoul 03181, Korea; 7Division of Gastroenterology and Hepatology, Department of Internal Medicine, Kangbuk Samsung Hospital, Sungkyunkwan University School of Medicine, Seoul 03181, Korea; choyk2004.cho@samsung.com; 8Department of Biochemistry, College of Medicine, Ewha Womans University, Seoul 07985, Korea; hyung@ewha.ac.kr; 9Division of Infectious Diseases, Department of Internal Medicine, Kangbuk Samsung Hospital, Sungkyunkwan University School of Medicine, Seoul 03181, Korea; hs.cheong@samsung.com

**Keywords:** hepatitis B virus, gut microbiome, *Megasphaera*

## Abstract

The changes in the gut microbiota of healthy hepatitis B virus (HBV) carriers, including asymptomatic and non-cirrhotic subjects, have been rarely scrutinized. From 1463 faecal samples in health examinees, in total 112 subjects, including 36 hepatitis B surface antigen (HBsAg)-positive and 76 control subjects, were included. Twenty-eight of 36 HBsAg-positive individuals (78%) showed normal alanine aminotransferase (ALT) levels (normal ALT group), whereas eight subjects exhibited elevated ALT levels (22%, high ALT group). By using 16S rRNA gene sequencing, the distance between normal and high ALT groups among HBsAg-positive subjects showed a significant separation after the pairwise comparison of weighted UniFrac distance (permutational analysis of variance *q*-value = 0.039), when compared with the distances to the control group. In comparison with the control group, the normal ALT group had *Anaerostipes* as a significant taxon that showed a positive association (Coefficient (*Coef.*) = 0.028, *q* = 0.039). *Desulfovibrio* (*Coef.* = 0.54, *q* = 0.014) and *Megasphaera* (*Coef.* = 1.41, *q* = 0.030) showed positive correlations, and *Acidaminococcus* (*Coef.* = −1.31, *q* = 4.15 × 10^−75^) exhibited a negative correlation with high ALT level. Gut microbial composition was different according to HBV-induced serum ALT levels, indicative of a potential link between gut and liver metabolism.

## 1. Introduction

Chronic hepatitis B virus (HBV) infection continues to be a major public health issue worldwide despite the availability of an effective vaccine and potent antiviral treatments [1]. Based on population estimates as per the United Nations Population Division for the year 2015, it was estimated that over 257 million individuals were hepatitis B surface antigen (HBsAg)-positive worldwide. In addition, the Western Pacific region including China, South Korea, Japan, the Philippines, and Vietnam accounted for nearly 50% of all chronic HBV infections. In South Korea, the prevalence of HBV carriers ranged from 8% to 10% in the 1980s and early 1990s; however, the overall prevalence has been maintained at 2.9% since 2013 after the launch of a universal vaccination program [2,3]. Although the prevalence of chronic HBV infection is decreasing, HBV infection is still a major aetiology of chronic liver diseases such as liver cirrhosis and hepatocellular carcinoma (HCC) in Korea [4].

Interactions between gut microbiota and liver diseases have received considerable attention in recent years [5]. Chronic HBV infection persists for a long time and is characterized with changes in the liver tissue, which may give rise to asymptomatic carriers and eventually progress to hepatic cirrhosis as well as to HCC. In previous studies, gut microbiota dysbiosis was observed in HBV-infected patients with liver fibrosis and cirrhosis [6,7]; this observation suggests that the gut microbiota may affect the progression of chronic hepatitis B to severe liver failure. Recent studies have investigated the relationship between the diversity and composition of human gut microbiota and HBV-induced chronic liver diseases, including asymptomatic carriage, chronic hepatitis B, and decompensated cirrhosis [8,9]. In comparison with healthy controls, patients with chronic hepatitis and cirrhosis exhibit remarkable alterations in the composition of *Bifidobacterium*. However, these observations have been reported in hospitalized patients with chronic HBV infection that may convey severe comorbidity. In terms of healthy HBV carriers, including asymptomatic and non-cirrhotic subjects who do not require hospitalization, the changes in the gut microbiota have been rarely scrutinized. In order to investigate the composition and diversity of the gut microbiota in healthy and asymptomatic HBV carriers, the present study included a study population that participated in health examination. Chronic HBV infection comprises several phases that are linked to the level of HBV replication and the strength and targets of the host immune response against the replicating HBV [10]. In particular, the immune reaction is considered as an important characteristic that determines the development and progression of HBV infection [11]. Therefore, we hypothesize that determination the serum alanine aminotransferase (ALT) level reflects hepatocyte damage associated with immune response against HBV replication and subsequently affects the gut microbiome. Categorizing asymptomatic carriers of HBV into those with high and normal ALT levels, we investigated the association of the gut microbiomes with the host immune response against HBV replication.

## 2. Materials and Methods

### 2.1. Study Subjects

The present analysis included all the study participants that underwent serologic testing for HBsAg with faecal samples for comprehensive health check-ups between June and September 2014 at Kangbuk Samsung Hospital Total Healthcare Screening Centres in Seoul, South Korea [12].

### 2.2. Inclusion and Exclusion Criteria

A case-control study was designed using the data from 1463 faecal samples from a cohort of participants enrolled in the Kangbuk Samsung Health Study during the study period [13]. In total, 45 subjects were detected positive for HBsAg among 1463 participants (3.1%). Thirty-nine subjects were retained after excluding participants with previous use of antibiotics, probiotics, or cholesterol-lowering medications (*n* = 2) and those with diabetes mellitus (*n* = 4); these subjects were age- and sex-matched to HBsAg-negative controls at 1:2 ratio (*n* = 78) with the same exclusion criteria. Liver cirrhosis subjects from HBsAg-positive (*n* = 3) and control (*n* = 2) groups were excluded, resulting in a total 112 subjects, including 36 HBsAg-positive and 76 control groups (Figure 1).

This study was approved by the Institutional Review Board of Kangbuk Samsung Hospital (KBSMC 2013-01-245-008, registered 23 December 2013). All study participants gave written informed consent to participate in the study. The present study was conducted according to the ethical guidelines of the World Medical Association Declaration of Helsinki.

### 2.3. Measurements

Data on medical history, medication use, family history, physical activity, alcohol consumption, smoking habits, and sociodemographic characteristics were collected through a self-administered, structured questionnaire. Anthropometric parameters and blood pressure were measured by trained staffs during health examination [14]. Body mass index (BMI) was calculated as weight (kg) over squared height (m^2^).

Blood specimens were collected after at least 10 h of fasting. Hepatitis B serologic testing was performed using electrochemiluminescent immunoassays (Modular E170; Roche Diagnostics, Tokyo, Japan). The presence of HBsAg at first testing was interpreted as an indication of chronic HBV infection. Serum biochemical parameters, including serum levels of glucose, ALT, and aspartate aminotransferase (AST), were assessed. Insulin resistance was evaluated using the homeostasis model assessment of insulin resistance (HOMA-IR) according to the following equation: fasting blood insulin (µU/mL) × fasting blood glucose (mg/dL)/405. Diabetes mellitus was defined as a fasting serum glucose level ≥ 126 mg/dL or current use of blood glucose-lowering agents. The levels of total cholesterol, low-density lipoprotein cholesterol (LDL-C), and high-density lipoprotein cholesterol (HDL-C) were directly measured with a homogeneous enzymatic colorimetric assay.

### 2.4. DNA Extraction and Sequence Data Generation

Faecal samples were immediately frozen after collection. 16S rRNA genes were extracted and amplified from the specimens using the MO-BIO PowerSoil DNA Isolation Kit (MO-BIO Laboratories, Carlsbad, CA, USA) according to the manufacturer’s instructions. Amplification and sequencing were performed in the same batch as previously described for the analysis of bacterial communities. The genomic DNA was amplified using fusion primers targeting 16S V3-V4 rRNA gene with indexing barcodes. All samples were pooled for sequencing on the Illumina Miseq platform according to the manufacturer’s specifications [15].

### 2.5. Sequence Analysis

Quality filtering, chimera removal, and de novo operational taxonomic unit (OTU) clustering were carried out using the UPARSE pipeline [16], which identifies highly accurate OTUs from amplicon sequencing data. The reads were dereplicated, sorted, and clustered into candidate OTUs by removing chimeric OTUs. Taxonomic assignment for OTUs was annotated by the Ribosomal Database Project reference (version 16) with an identity threshold of 97% using the UTAX command in the UPARSE pipeline. The OTU table with taxonomic assignments was exported to QIIME2 software (version 2017.10; http://qiime.org). A total of 2,119,047 reads/204 OTUs, with a mean of 18,925 (SD = 11,735) sequences per faecal sample, were included in the QIIME analysis. For diversity analysis, we rarefied the data to 2500 sequences per sample. Sample biodiversity (i.e., alpha diversity) was estimated according to different microbial diversity metrics (i.e., Faith’s phylogenetic distance, evenness, Shannon index, observed OTUs). To determine the dissimilarity between samples (i.e., beta diversity), weighted and unweighted UniFrac distance matrix were used, and principal coordinates analysis (PCoA) was performed on the distance matrix [17]. Permutational analysis of variance (PERMANOVA) for pairwise comparison of the distance matrix was calculated with 999 Monte Carlo permutation and Bonferroni multiple correction.

### 2.6. Statistical Analysis

Basic statistical analyses for sample characteristics were performed using SPSS (version 18.0.0, SPSS Inc., Chicago, IL, USA). Relationships between the abundance of one or more taxa and subgroups based on ALT levels (control, normal ALT, and high ALT) were examined. Multivariate association with linear model (MaAsLin) packages (http://huttenhower.sph.harvard.edu/maaslin) [18] of RStudio (version 0.98.983, RStudio Inc., Boston, MA, USA) was used to detect associations between microbiome composition and subgroups while considering the effects of confounders in the study population (i.e., age, sex, and BMI). The minimum relative abundance below 0.0001 was removed, and a feature of “at least 10% of samples” was included for statistic calculation. All analyses in MaAsLin were performed using the default options. The resulting *p*-values were corrected for multiple comparisons on each phylogenetic level using Benjamini–Hochberg correction (false discovery rate (FDR)). A value of *q* <0.05 was considered statistically significant.

## 3. Results

### 3.1. Baseline Characteristics of the Study Population

The present analysis included a total of 112 subjects, including 36 HBsAg-positive and 76 control subjects. Twenty-eight HBsAg-positive individuals showed normal ALT levels and were considered as the normal ALT group (28/36, 78%), whereas eight subjects with elevated ALT levels (8/36, 22%) were categorized as the high ALT group. The baseline characteristics of the study population are described in Table 1. No difference was observed in the mean of age among control, normal ALT, and high ALT groups. Participants with high ALT levels were more likely to be male and obese, when compared with those from the control and normal ALT groups (Table 1). HOMA-IR was positively but marginally associated with high ALT level (*p* = 0.066).

### 3.2. Microbial Diversity of the Gut Microbiota in Control, Normal ALT, and High ALT Groups

We analysed the gut microbiota from the faecal samples of control and HBV carriers with 16S rRNA gene sequencing. The relative abundance at the phylum and genus level was different between each group (Appendix A); however, no statistically significant result was observed in terms of biodiversity (alpha-diversity, Appendix A). Pairwise comparison revealed the reduction in the evenness in high ALT group, when compared with the normal ALT group, but no statistical significance was reported (*p* = 0.036, *q* = 0.109) (Figure 2). We performed PCoA using UniFrac to find clusters of similar groups (Figure 3A). We failed to observe any distinct clustering among control, normal, and high ALT groups in PCoA plot. However, UniFrac-based PCoA revealed that the microbiota in the high ALT group assumed a phylogenetically similar composition. In comparison to the distances to the control group, the distance between the normal and high ALT groups among HBsAg-positive subjects showed a significant separation in the pairwise comparison of weighted UniFrac distance (PERMANOVA *q* = 0.039) (Figure 3B, Appendix A). This observation indicates that the microbial community composition of subjects from the normal ALT group was rather similar to that of the subjects from the control group but significantly different from that of the subjects from the high ALT group.

### 3.3. Taxonomic Comparison between Control, Normal ALT, and High ALT Groups

To verify the change in the microbial components, we performed a multivariate regression model, MaAsLin, adjusted by age, sex, and BMI. This analysis of relative abundances at the phylum and genus levels revealed significant results only at the genus level (Figure 4). In comparison with the control group, the normal ALT group among HBsAg-positive subjects showed *Anaerostipes* as the significant taxon, which had positive association (Coefficient (*Coef.*) = 0.028, *q* = 0.039). In comparison between normal and high ALT groups, *Desulfovibrio* (*Coef.* = 0.54, *q* = 0.014) and *Megasphaera* (*Coef.* = 1.41, *q* = 0.030) showed positive correlations with high ALT group, while *Acidaminococcus* (*Coef.* = −1.31, *q* = 4.15 × 10^−75^), which is the same Negativicutes as *Megasphaera*, showed a negative correlation with high ALT group.

## 4. Discussion

In this present study, we revealed the altered diversity and composition of the gut microbiota of subjects with chronic HBV infection, when compared with the HBsAg-negative controls. We showed that the microbial structure significantly differed within the predefined subgroups (HBsAg-negative control and HBsAg-positive normal ALT and high ALT groups). The distinct patterns within the relative abundance at the genus level were associated with the stage of liver disease, defined through the ALT level. Overall, this study demonstrates the possible influence of the immune reaction to chronic HBV infection on gut microbiota. Recent studies have investigated the relationship between the diversity and composition of human gut microbiota and HBV-induced chronic liver diseases. Gut microbiota dysbiosis was noted in HBV-infected patients with liver fibrosis and cirrhosis [6,7,19], emphasizing the pathogenic role of gut dysbiosis in the progression of liver diseases and liver failure. According to Lu et al., the *Bifidobacteriaceae*/*Enterobacteriaceae* (B/E) ratio was significantly decreased in patients with HBV infection, indicative of detractive microbial colonization resistance in the bowel, especially in those with cirrhosis [8]. In the present analysis with asymptomatic and non-cirrhotic HBV carriers and healthy controls, however, the B/E ratio was not different between the predefined subgroups. Aside from liver cirrhosis, the effect on the gut microbiota of asymptomatic and non-cirrhotic HBV carriers was rarely investigated. In the present case-control study, we aimed to identify the differences in the human gut microbiota associated with chronic HBV infection without cirrhosis.

There exists a strong relationship between the liver and the gut via portal system, which receives blood from the gut; the intestinal blood content activates liver function. The liver, in turn, affects intestinal functions through bile secretion in the intestinal lumen. The present study demonstrates that *Megasphaera* genus belonging to the phylum *Firmicutes* was relatively abundant in high ALT group than in normal ALT group. Studies on *Megasphaera* spp. have suggested that this is an important member of the rumen microbiome and exerts beneficial effects on the host [20]. It is thought to exhibit a positive role in the human gut by producing short-chain fatty acids (SCFAs) such as propionate, which is a major product of lactate fermentation [21]. In this study, the positive correlation with *Megasphaera* genus was observed in high ALP group among HBsAg-positive subjects; this observation may be associated with the link between HBV and liver metabolism, involving bile acid synthesis and cholesterol provision [22]. HBV replication in hepatocyte could lead to a host immune reaction characterized with elevated ALT levels, and may promote changes in bile acid and lipid metabolism in the liver and gut; in turn, the relative abundance of *Megasphaera* spp. may be increased in the high ALT group, considering its bile-resistant trait and host-specific adaptation [21]. *Anaerostipes*, as butyrate-producing bacteria [23], were found to be more abundant in the faeces of HBsAg-positive subjects with normal ALT relative to faeces of HBsAg-negative controls. Butyrate, an SCFA similar to propionate, is a main end-product of intestinal microbial fermentation of dietary fibres and has anti-carcinogenic and anti-inflammatory potentials. Butyrate may affect the intestinal barrier and play a role in satiety and oxidative stress [24]. The present study identified the positive association of lactate-utilizing bacteria such as *Anaerostipes* in normal ALT group among HBsAg-positive subjects. The analysis based on the ALT levels in HBsAg-positive subjects revealed the differences in the gut microbial composition based on HBV-induced serum ALT levels and highlighted the positive association between elevated ALT levels and relative abundance of *Megasphaera*; this observation indicates the potential link between gut and liver metabolism promoted by HBV replication. The natural course of chronic HBV infection includes immune tolerance, immune clearance, and residual or inactive phase with different immunologic and virological features. Therefore, the role of HBV infection in the altered gut microbiota may vary in different phases [10]. To establish if HBV infection affects the diversity and composition of gut microbiome differently in various phases, additional studies using HBV DNA, hepatitis B e-antigen (HBeAg), and antibody to HBeAg are warranted.

The prevalence of non-alcoholic fatty liver disease (NAFLD) is increasing worldwide. It is the most common cause of chronic liver disease among children and adolescents and is as a major public health problem [25]. In our previous study, HBV infection was shown to exert a protective role in the development of NAFLD, probably owing to the effects of HBV on lipid metabolism as well as other undetermined factors [26]. The beneficial effects of chronic HBV infection on lipid profiles were identified in previous reports such as those demonstrating the lower prevalence of metabolic syndrome (MetS) in chronic HBV-infected patients [27,28,29]. Based on the paradoxically beneficial effect of chronic HBV infection on metabolic diseases in healthy HBV carriers, this study emphasizes the changes in the gut microbiota induced by chronic HBV infection, which may play a key role in the association between HBsAg positivity and metabolic diseases. Herein, we found that chronic HBV infection affected the relative abundance of butyrate-producing bacteria in the normal ALT group. The gut microbial composition changed according to HBV-induced serum ALT levels, suggesting that the liver injury promoted by HBV replication may affect the lactate-using bacteria that are known to produce SCFAs and enhance the metabolic health. The mechanisms underlying the influence of HBV on microbial composition and how these changes affect the development of metabolic diseases are not well understood. The presence of a gut–vascular barrier, which controls the antigen access to the portal vein in the gut–liver axis, may affect the interaction between the gut microbiota and HBV-induced metabolic diseases [30]. As the gut microbiota may modulate the host metabolic phenotypes, further studies are required to identify the effect of gut microbiota on the metabolic health of the HBV-infected host.

### Limitation

We note that our study has several limitations. First, the results failed to demonstrate HBV DNA level and serologic results of HBeAg and HBsAg antibody as useful parameters to evaluate HBV replication. Therefore, the phases of chronic HBV infection could not be specified; this limitation may affect the microbial community. To define immunologic responses to HBV infection, we categorized the subjects into two subgroups (high and normal ALT) to correlate the biochemical response with DNA response. Second, this study was predominated by male subjects. Although we selected a control group that was age- and sex-matched with HBV carriers, sex disparity was evident in the high ALT group among HBsAg-positive subjects. In order to control for sex disparity, we adjusted for confounders in the MaAsLin model. Third, the effects of potential confounders may exist, such as dietary information and usage of antiviral therapy, proton pump inhibitors and other medications which we have not noticed. Fourth, our findings in healthy Korean men and women from HBV endemic area may not be generalized to other populations. Fifth, data regarding metabolomes were not provided in the present analysis, which is important for an interactive host–microbiota metabolic signaling. Instead, fatty liver defined by ultrasonography was used as a surrogate of gut barrier function. Finally, we have a technical limitation from 16S amplicon-based sequencing data that may introduce biases through polymerase chain reaction amplification steps and determine only the genus level at the maximum.

## 5. Conclusions

Despite the limitations, our study results have an important finding that demonstrates the association between HBV infection and composition of gut microbiota. Gut microbial composition was different according to HBV-induced serum ALT levels, indicative of the potential link between gut and liver metabolism promoted by HBV replications. These results may highlight the possible influence of chronic HBV infection on the metabolic health of the host.

## Figures and Tables

**Figure 1 jcm-08-00173-f001:**
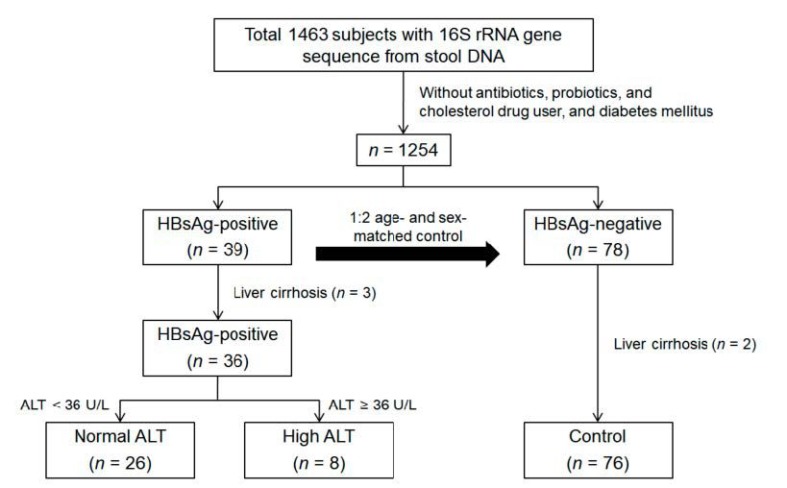
Flow chart of the study subjects. ALT: alanine aminotransferase; HBsAg: hepatitis B surface antigen.

**Figure 2 jcm-08-00173-f002:**
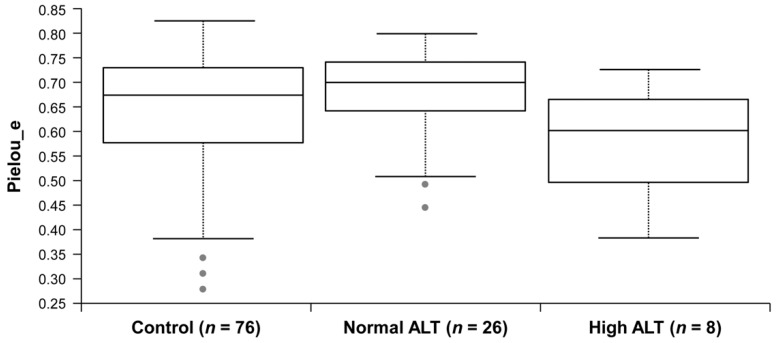
Comparison of Pielou’s Evenness across sample groups. Control versus normal ALT (*p* = 0.238, *q* = 0.238), control versus high ALT (*p* = 0.152, *q* = 0.228), normal ALT versus high ALT (*p* = 0.036, *q* = 0.109). The values of *q* were calculated with Benjamini–Hochberg correction (FDR).

**Figure 3 jcm-08-00173-f003:**
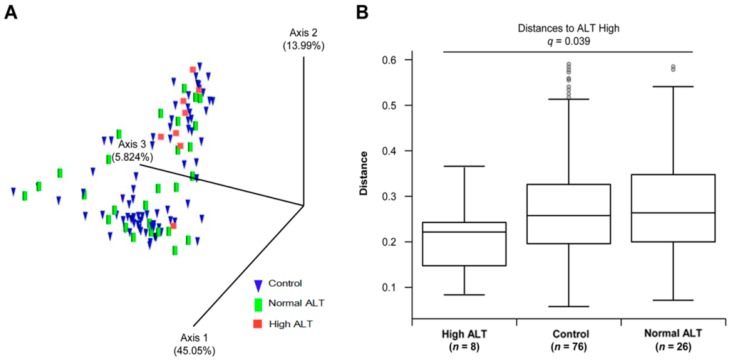
Comparison of beta-diversity between sample groups. (**A**) Principal coordinates analysis (PCoA) plots according to weighted UniFrac. Control: blue, normal ALT: green, high ALT: red. (**B**) Comparison of weighted UniFrac distant matrices between each group. Normal ALT distance to high ALT was significant (*q* = 0.039). The values of *q* were calculated by Benjamini–Hochberg correction (FDR).

**Figure 4 jcm-08-00173-f004:**
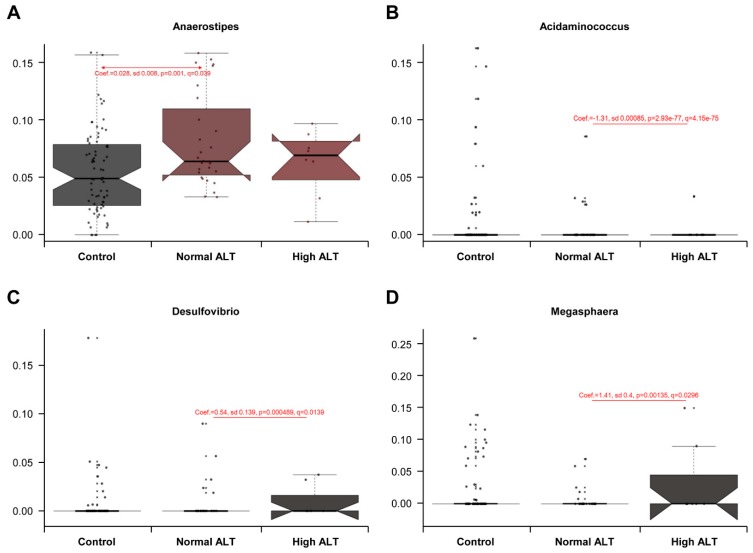
Significant associations between three groups and microbial taxa, *Anaerostipes* (**A**), *Acidaminococcus* (**B**), *Desulfovibrio* (**C**), and *Megasphaera* (**D**). The coefficient (*Coef.*) and *q* value were shown in each plot. The values of *q* were determined with MaAsLin analysis.

**Table 1 jcm-08-00173-t001:** Characteristics of the studied groups.

Demographic Characteristics	HBsAg-Negative Control (*n* = 76)	HBsAg-Positive Normal ALT (*n* = 28)	HBsAg-Positive High ALT (*n* = 8)	*p*-Value ^a^
Male (number)	53 (70%)	17 (37%)	8 (100%)	0.103
Age (years)	45.07 ± 6.74	45.82 ± 7.03	49.00 ±4.28	0.460
BMI (kg/m^2^)	23.44 ± 3.01	23.99 ± 2.72	26.89 ± 2.68	0.007
Fat percent (%)	24.41 ± 5.93	26.86 ± 6.43	26.24 ± 5.65	0.174
Systolic BP (mmHg)	110.39 ± 13.41	109.21 ± 13.46	117.50 ± 15.51	0.308
Diastolic BP (mmHg)	72.03 ± 10.61	71.36 ± 10.76	73.88 ± 10.52	0.839
Glucose (mg/dL)	94.99 ± 8.73	91.57 ± 7.14	94.13 ± 8.17	0.184
Total-C (mg/dL)	204.28 ± 36.69	191.54 ± 38.05	202.13 ± 40.83	0.306
HDL-C (mg/dL)	55.41 ± 15.00	55.18 ± 11.89	64.63 ± 12.86	0.207
LDL cholesterol (mg/dL)	128.25 ± 32.70	114.79 ± 33.01	121.63 ± 35.21	0.181
Triglycerides (mg/dL)	128.76 ± 75.03	115.46 ± 70.44	86.25 ± 28.97	0.241
Total Bilirubin (mg/dL)	0.83 ± 0.43	0.83 ± 0.36	0.86 ± 0.21	0.982
AST (IU/L)	23.39 ± 15.42	21.86 ± 7.19	39.25 ± 8.84	0.005
ALT (IU/L)	25.61 ± 26.21	20.36 ± 7.94	51.88 ± 17.07	0.003
WBC (×10^3^/mm^3^)	5.93 ± 1.54	5.52 ± 1.19	5.11 ± 0.96	0.181
Platelet (×10^3^/mm^3^)	250.09 ± 47.86	222.86 ± 45.37	182.38 ± 44.76	<0.001
HOMA-IR	1.35 ± 0.80	1.28 ± 0.54	1.96 ± 0.67	0.066
Fatty liver ^b^ (%)	39 (30/76)	18 (5/28)	25 (2/8)	0.102

The values are expressed as the mean ± standard deviation or frequency. BMI: body mass index, BP: blood pressure, HDL: high density lipoprotein, LDL: low-density lipoprotein, AST: aspartate aminotransferase, ALT: alanine aminotransferase, WBC: white blood cell, HOMA-IR: homeostasis model assessment of insulin resistance. ^a^
*p*-value for difference between groups from Kruskal-Wallis test for continuous variables and *X*^2^ test for categorical variables. ^b^ Fatty liver was diagnosed with ultrasonography.

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
