# Peer review of "Alterations of the Gut Microbiome in Chronic Hepatitis B Virus Infection Associated with Alanine Aminotransferase Level"

_jcm, 2019, doi:10.3390/jcm8020173_

Reviewer 1 Report

This manuscript describes a study comparing several biomarkers as well as microbiota composition (by 16S sequencing of fecal samples) of asymptomatic HBV carriers and age-matched controls. 

The authors describe significant differences in the genera of the microbiota between the groups, primarily between the HBV carriers with and without a biomarker of HBV infection (the enzyme ALT). However, as the authors themselves attest, the conclusions that can be drawn from these results are circumscribed somewhat by a dramatic sex imbalance between the normal-alt (primarily female) and high-alt (entirely male) HBV+ groups. The observation that disease in HBV+ males seem to progress more rapidly may be valuable on its own (the authors do not describe the literature on that point.) but it tempers enthusiasm for the other results.

In general, I believe this manuscript could be improved by greater discussion of these results in the context of prior studies on sex differences in HBV infection. I note three specific points below:

Study participants-

Out of 45 HBV+ participants originally identified, only 6 were ruled out because of prior use of antibiotics. Perhaps the authors could clarify; it seems unlikely that 39 out of 45 had no history of antibiotic usage whatsoever, perhaps the 6 were ruled out for recent antibiotic use?

Sex-

As noted by the authors, the differences in sex ratio between the groups are potentially important, particularly between the normal-ALT and high-ALT groups. Sex differences in the microbiome are well documented in the literature which might weigh on the results shown in Figure 3B. Perhaps the authors could explain the sex-adjustment in the MaAsLin model?

B/E ratio –

the Authors mention the results from Lu et al but do not provide results from their own data. Even if they saw no significant differences, it is worth mentioning that.

Author Response

1. Study participants-

Out of 45 HBV+ participants originally identified, only 6 were ruled out because of prior use of antibiotics. Perhaps the authors could clarify; it seems unlikely that 39 out of 45 had no history of antibiotic usage whatsoever, perhaps the 6 were ruled out for recent antibiotic use?

Response: We agree with the reviewer’s comments. As the reviewer recommended, we provided the excluded number in the revised manuscript.

ð  Thirty-nine subjects were retained after excluding participants with previous use of antibiotics, probiotics, or cholesterol-lowering medications (n=2) and those with diabetes mellitus (n=4) (page 4, lines 97-99)

2. Sex-

As noted by the authors, the differences in sex ratio between the groups are potentially important, particularly between the normal-ALT and high-ALT groups. Sex differences in the microbiome are well documented in the literature which might weigh on the results shown in Figure 3B. Perhaps the authors could explain the sex-adjustment in the MaAsLin model?

Response: We would like to thank the reviewer for their insightful comments. As the reviewer recommended, we added this to the limitation section of the revised manuscript as follows:

ð  Second, this study was predominated by male subjects. Although we selected a control group that was age- and sex-matched with HBV carriers, sex disparity was evident in the high ALT group among HBsAg-positive subjects. In order to control for sex disparity, we adjusted for confounders in the MaAsLin model (page 17, lines 321-325).

3. B/E ratio –

the Authors mention the results from Lu et al but do not provide results from their own data. Even if they saw no significant differences, it is worth mentioning that.

Response:As the reviewer recommended, we compared B/E ratio between the predefined subgroups (HBsAg-negative control and HBsAg-positive normal ALT and high ALT groups), revealing no significant differences between groups. We added following sentences in the revised manuscript.

ð  In the present analysis with asymptomatic and non-cirrhotic HBV carriers and healthy controls, however, B/E ratio was not different between the predefined subgroups (page 14, lines 253-255).

Reviewer 2 Report

The study is important and interesting, however authors must consider that many factors can influence gut microbiota and also that metabolic function of microbiota can play crucial role in diseases development. Including the following points will improve significantly this manuscript.

1.Line 84 - Authors should define the aim of the study and hypothesis, which they want to check

2. Line 97 – Authors should provide information about other drugs, which can influence gut microbiota, especially PPIs, NSAIDs and psychotropic agents.

3. Was the diet included in the process of matching? If not this issue should be detailed discussed and considered as the limitation.

4. It was significant difference of BMI between high ALT and normal ALT groups. Because BMI can have significant influence on gut microbiota this factor must be included in the statistiical analysis of data.

5. Why did authors not make the analysis of association between high AST level and microbiota?

6. Authors should provide the analysis of metagenome functional content from marker gene, using for example PICRUSt (Phylogenetic Investigation of Communities by Reconstruction of Unobserved States).

7. Lack of metabolomic analysis (for example SCFA) is the important study limitation as well as lack of analysis of gut-barrier function. Maybe authors have results of the fatty liver stage in ultrasound examination, which can be considered as a surrogate of gut barrier function. These limitations must be included in the study.

8. Please include in discussion the problem of gut-vascular barrier, which plays an important role in the gut-liver axis (Spadoni I, Zagato E, Bertocchi A, Paolinelli R, Hot E, Di Sabatino A, Caprioli F, Bottiglieri L, Oldani A, Viale G, Penna G, Dejana E, Rescigno M. A gut-vascular barrier controls the systemic dissemination of bacteria. Science.2015;350(6262):830-4. doi: 10.1126/science.aad0135).

Author Response

Reviewer #2

The study is important and interesting, however authors must consider that many factors can influence gut microbiota and also that metabolic function of microbiota can play crucial role in diseases development. Including the following points will improve significantly this manuscript. 

Response: Thank you for your supportive comments. We have addressed your comments and suggestions below.

1. Line 84 - Authors should define the aim of the study and hypothesis, which they want to check

Response: Thank you for this comment. We defined the aim of the study and hypothesis in the revised manuscript

ð  Therefore, we hypothesize that determination the serum alanine aminotransferase (ALT) level reflects hepatocyte damage associated with immune response against HBV replication and subsequently affects the gut microbiome. Categorizing asymptomatic carriers of HBV into those with high and normal ALT levels, we investigated the association of the gut microbiomes with the host immune response against HBV replication (page 3, lines 79-84).

2. Line 97 – Authors should provide information about other drugs, which can influence gut microbiota, especially PPIs, NSAIDs and psychotropic agents.

Response: Thank you for pointing this out. At the time of enrollment, we excluded subjects who had took medication which could affect gut microbiota such as antibiotics, probiotics and cholesterol lowering medication (methods, lines 97-99). The information about PPI usage, however, was not investigated. We added this to the limitation section of the revised manuscript as follows:

ð  Third, the effects of potential confounders may exist such as dietary information and usage of antiviral therapy, proton pump inhibitors and other medications which we have not noticed (page 17, lines 325-327).

3. Was the diet included in the process of matching? If not this issue should be detailed discussed and considered as the limitation.

Response: We appreciate the reviewer’s comments. As the reviewer suggested, we have added this point in the limitation section of the revised manuscript. Please refer to the response for the second question.

4. It was significant difference of BMI between high ALT and normal ALT groups. Because BMI can have significant influence on gut microbiota this factor must be included in the statistiical analysis of data.

Response: We agree with the reviewer’s comments that BMI has a significant influence on the gut microbiome. Therefore, we performed a multivariate regression model (MaAsLin) to adjust for age, sex and BMI (page 12, lines 222-223).

5. Why did authors not make the analysis of association between high AST level and microbiota?

Response: Determination of serum ALT level is important for starting antiviral treatment as well as for follow-up of patients with chronic HBV infection, instead of serum AST level (reference: Sarin S.K.; et al. Asian-Pacific clinical practice guidelines on the management of hepatitis B: a 2015 update. Hepatol Int. 2015, 10, 1-98.)

6. Authors should provide the analysis of metagenome functional content from marker gene, using for example PICRUSt (Phylogenetic Investigation of Communities by Reconstruction of Unobserved States).

Response: Thank you for your suggestion. In our analysis, we used the UPARSE pipeline to carry out quality filtering, chimera removal, and de novo operational taxonomic unit (OTU) clustering. Since PICRUSt requires only closed-reference OTU picking method, it’s difficult to apply PICRUSt for our data. So we tried “Piphillin”tool instead of PICRUSt, regarded as an improved prediction than PICRUSt. Above all, Piphillin has no restriction of the database for input data (https://journals.plos.org/plosone/article?id=10.1371/journal.pone.0166104, Table 2). We performed the web version of Piphillin (http://secondgenome.com/Piphillin), which gave KEGG orthology pathway information from OTU abundance table. Unfortunately, the result of Piphillin did not show significant differences among groups (nothing significant by FDR multiple correction (q<0.25), supplemental excel file, 1-way ANOVA by Kruskal-Walis test for three groups, and by Mann-Whitney U test for two groups). Eventually, it’s difficult to predict traits by comparing short 16S reads to sparse reference databases, as there are some debates said “PICRUSt approach is skeptical that reliable predictions are possible” (reference:  https://www.drive5.com/usearch/manual/cmd_closed_ref.html). 

7. Lack of metabolomic analysis (for example SCFA) is the important study limitation as well as lack of analysis of gut-barrier function. Maybe authors have results of the fatty liver stage in ultrasound examination, which can be considered as a surrogate of gut barrier function. These limitations must be included in the study.

Response: We agree with the reviewer’s comments. We added this point in the limitation.

ð  Fifth, data regarding metabolomes were not provided in the present analysis, which is important for an interactive host-microbiota metabolic signaling. Instead, fatty liver defined by ultrasonography was used as a surrogate of gut barrier function (page 17, line 328-page 18, line 331)

8. Please include in discussion the problem of gut-vascular barrier, which plays an important role in the gut-liver axis (Spadoni I, Zagato E, Bertocchi A, Paolinelli R, Hot E, Di Sabatino A, Caprioli F, Bottiglieri L, Oldani A, Viale G, Penna G, Dejana E, Rescigno M. A gut-vascular barrier controls the systemic dissemination of bacteria. Science.2015;350(6262):830-4. doi: 10.1126/science.aad0135).

Response: Thank you for the reviewer’s suggestions. We added the following sentences in the discussion section of the revised manuscript.

ð  The presence of a gut-vascular barrier, which controls the antigen access to the portal vein in the gut-liver axis, may affect the interaction between the gut microbiota and HBV-induced metabolic diseases (page 17, lines 309-313).

ð  30. Spadoni I.; Zagato E.; Bertocchi A.; Paolinelli R.; Hot E.;Di Sabatino A.; Caprioli F.; Bottiglieri L.; Oldani A.; Viale G.; Penna G.; Dejana E.; Rescigno M. A gut-vascular barrier controls the systemic dissemination of bacteria. Science. 2015, 350, 830-834.

Round  2

Reviewer 2 Report

Dear Authors,

Thank you for explanations and considering my suggestions. I hope that manusript was significantly improved.